# SARS-CoV-2 Infection Prompts IL-1β-Mediated Inflammation and Reduces IFN-λ Expression in Human Lung Tissue

**DOI:** 10.3390/pathogens11111390

**Published:** 2022-11-21

**Authors:** Bianca Vezzani, Margherita Neri, Stefano D’Errico, Alberto Papi, Marco Contoli, Carlotta Giorgi

**Affiliations:** 1Department of Medical Sciences, Section of Experimental Medicine, University of Ferrara, 44121 Ferrara, Italy; 2Laboratory of Technologies for Advanced Therapy (LTTA), Technopole of Ferrara, 44121 Ferrara, Italy; 3Department of Medical Sciences, Section of Public Health Medicine, University of Ferrara, 44121 Ferrara, Italy; 4Department of Medicine, Surgery and Health, University of Trieste, 34149 Trieste, Italy; 5Respiratory Medicine, Department of Translational Medicine, University of Ferrara, 44121 Ferrara, Italy

**Keywords:** SARS-CoV-2, COVID-19, IFN-λ, IL-1β, IL-6, lungs

## Abstract

Two years after its spreading, the severe acute respiratory syndrome coronavirus 2 (SARS-CoV-2) is still responsible for more than 2000 deaths per day worldwide, despite vaccines and monoclonal antibody countermeasures. Therefore, there is a need to understand the immune–inflammatory pathways that prompt the manifestation of the disease to identify a novel potential target for pharmacological intervention. In this context, the characterization of the main players in the SARS-CoV-2-induced cytokine storm is mandatory. To date, the most characterized have been IL-6 and the class I and II interferons, while less is known about the proinflammatory cytokine IL-1β and class III interferons. Here, we report a preliminary study aimed at the characterization of the lung inflammatory context in COVID-19 patients, with a special focus on IFN-λ and IL-1β. By investigating IFN and inflammatory cytokine patterns by IHC in 10 deceased patients due to COVID-19 infection, compared to 10 control subjects, we reveal that while IFN-β production was increased in COVID-19 patients, IFN-λ was almost abolished. At the same time, the levels of IL-1β were dramatically improved, while IL-6 lung levels seem to be unaffected by the infection. Our findings highlight a central role of IL-1β in prompting lung inflammation after SARS-CoV-2 infection. Together, we show that IFN-λ is negatively affected by viral infection, supporting the idea that IFN-λ administration together with the pharmaceutical blockage of IL-1β represents a promising approach to revert the COVID-19-induced cytokine storm.

## 1. Introduction

It is now well recognized that COVID-19 progression is deeply associated with inflammation and subjects’ immune response, highlighting that alterations in immunity and cytokine production can be implicated in disease severity and outcome. Therefore, there is a deep need to understand the main players of SARS-CoV-2-induced cytokine storm.

Cytokines play an important role in modulating the immune system and inflammation during viral infection. However, if a fast and coordinated innate immune response is fundamental in repressing viral infection, an exacerbating inflammation, called a “cytokine storm”, may cause tissue damage. Furthermore, the inflammatory burst that follows the infections can negatively interplay with immune responses, leading to increased severity of viral infections [1,2].

As for other coronavirus (CoV) infections, COVID-19 is characterized by intense stimulation of the innate and inflammatory responses, which results in the activation of several immune–inflammatory pathways, including the Nod-like receptor family, pyrin domain-containing 3 (NLRP3) inflammasome pathway with the consequent production of proinflammatory cytokines such as interleukin (IL-) 1β [3,4]. Recent studies reported increased blood levels of IL-1β in COVID-19 patients compared to healthy subjects [5,6]. Similarly, it has been reported that TNF-α, IFN-γ, IL-2, IL-4, IL-6, and IL-10 are also highly expressed in the plasma of COVID-19 patients; moreover, IL-6 and IL-10 are significantly increased in the serum of patients with critical conditions, compared to the moderate ones [7]. However, a transcriptional study performed on lung tissue has shown that the expression of IL-6 was not affected by SARS-CoV-2 infection [8], raising a question about the role of IL-6 therapies in COVID-19. Furthermore, randomized trials with anti-IL6 treatments in COVID-19 returned mixed results in terms of clinical benefit with these interventions.

Interferons (IFNs) are pivotal components of the early defense system against viral infections. CoV—and specifically SARS-CoV-2—infections have been shown to poorly induce IFNs production, a mechanism of possible viral immune escape [9,10]. Interestingly, several in vivo studies showed delayed and/or impaired type I (IFN-α and IFN-β) and type III (IFN-λ) interferon production in blood samples or nasopharyngeal swabs of COVID-19 patients particularly linked to the more severe manifestation of the disease [1,2,11,12,13]. However, recent studies failed to identify blood interferon levels as a marker of severity and clinical status in COVID-19 patients [14] and other studies showed that IFNs are overrepresented in the lower airways of patients with severe COVID-19 [15]. Furthermore, while subcutaneous administration of peginterferon-λ accelerated viral decline in outpatients with COVID-16 [16], administration of type I IFNs, particularly in the late and severe phase of the disease, leads to conflicting results and possible side effects [17].

Because of the heterogeneity of the underlying immune–inflammatory mechanisms that pave the way to the manifestation of COVID-19 and because of the need of identifying biological targets for potential interventions, we evaluated postmortem lung specimens (i.e., the site of SARS-CoV-2 infection) of severe COVID-19 patients and controls for the expression of immune-inflammatory biomarkers, including IL-1β, IL-6, IFN-β, and IFN- λ.

## 2. Materials and Methods

### 2.1. Case Selection

This study was performed by using human postmortem lung samples, collected during autopsies ordered by the prosecutor and used after the end of the investigations. Ethical approval was obtained by the competent Ethic Committee (CE-AVEC), with the approval N 342/2020/Oss/AOUFeO. A total of 10 COVID-19-positive subjects were included (Group 1), and as a control group, we selected 10 subjects who died from polytrauma or gunshot head without the presence of concomitant known infectious lung diseases (Group 2). Samples were anonymized by assigning them an alphanumeric code.

### 2.2. Autopsies and Tissue Processing

The autopsies and the sample collection were executed between two and seven days after the death, of all the enrolled subjects. The autopsies of patients who died because of severe COVID-19 were performed at the University of Trieste (Trieste, Italy), while the autopsies of the control cases were performed at the University of Ferrara (Ferrara, Italy). Autopsies were performed in infection isolation rooms according to Italian law. Histological samples obtained after the autopsy were fixed in 10% buffered formalin for 48 h and then included in paraffin for further processing.

### 2.3. Histological and Immunohistochemical Analysis

Paraffin-embedded lung sections (5 µm thick) were evaluated for the expression of multiple cytokines, such as IL-1β, IL-6, IFN-β, and IFN-λ. The dilution of antibodies and pretreatments for antigen retrieval were as follows: IL-1β (sc-32294, Santa Cruz Biotechnology, Santa Cruz, CA, USA), retrieval 0.1 M citrate buffer pH 6.00, 8 min at 100 °C, dilution 1:200 for 2 h at 20 °C; IL-6 (sc-130326, Santa Cruz Biotechnology), retrieval Proteinase K, 15 min at 20 °C, dilution 1:500 for 2 h at 20 °C; IFN-β (APA222P001, Cloud Clone Corp., Katy, TX, USA), retrieval EDTA buffer 0.5 M pH 8.00, 8 min at 100 °C, dilution 1:50 o.n. at 4 °C; IFN-λ (IL28A) (GTX31148, Genetex, Irvine, CA, USA), retrieval EDTA buffer 0.5 M pH 8.00, 8 min at 100 °C, dilution 1:500 o.n. at 4 °C.

Primary antibodies were detected using a biotinylated secondary antibody and horseradish peroxidase-conjugated streptavidin (4plus HRP Universal Detection, Biocare Medical, Pacheco, CA, USA). 3,3′-Diaminobenzidine (DAB, Biocare Medical, Pacheco, CA, USA) and H_2_O_2_ (Betazoid DAB Chromogen Kit, Biocare Medical, CA, USA) were used as the chromogen and substrate, respectively. Subsequent counterstaining with hematoxylin allowed for the visualization of nuclei.

The samples were then analyzed with a semiquantitative evaluation. Each slide was evaluated by 2 different investigators at ×40 magnification. The intensity of the overall immunopositivity was assessed semiquantitatively and expressed on a scale of 0–8 as follows: 0, no immunoreactivity (0%); 1, basal immunopositivity (5%); 2–3, isolated immunopositivity (15%); 4, mild immunopositivity (30%); 5–6, diffuse immunopositivity (50%), 7, highly diffuse immunopositivity (75%); 8, widespread immunopositivity (>90%). In cases of divergent scores, a third investigator was consulted.

### 2.4. Statistical Analysis

Statistical analysis was performed using the software Prism, firstly by identifying the distribution type and then by comparing the determined expression levels of each cytokine between the two groups (CTRL vs. COVID-19) using Student’s *t* test with Welch correction in case of normal distribution, or Mann–Whitney test in case of non-normal distribution.

## 3. Results

### 3.1. Clinical Overview of the Enrolled Subjects

Group 1 included 10 COVID-19 patients affected by the original form of SARS-CoV-2, with no vaccination. Group 1 patients were aged between 30 to 85 years, all males, hospitalized for periods between 5 and 30 days, and died between the 30th of March 2020 and the 30th of March 2021. COVID-19 patients were enrolled among those with no serious comorbidity described in hospital documentation. Specifically, this group included two obese patients, three affected by diabetes mellitus, and six with arterial hypertension. Group 2, as the control group, included 10 male subjects aged between 15 to 85 years, who died from head trauma caused by gunshot or traffic accident without the presence of concomitant known infectious lung diseases.

### 3.2. COVID-19 Patients Have Increased IL-1β Expression in Lung Tissue

We performed immunohistochemical analysis on lung samples to estimate the expression of multiple pro-inflammatory cytokines that can be potentially involved in the pathogenesis of COVID-19-associated pneumonia. We observed that IL-6 expression in lung tissue belonging to COVID-19 patients was comparable to what was observed in control subjects (Figure 1A,B and Appendix A).

Interestingly, IL-1β expression was triggered by SARS-CoV-2 infection. In fact, IL-1β levels present in COVID-19 patients were statistically higher compared to the control group (Figure 1C,D and Appendix A).

### 3.3. IFN-λ Is Reduced in COVID-19 Patients

Surprisingly, the analysis of INF-β expression revealed an increased positivity in COVID-19 patients, compared to what was observed in control subjects (Figure 2A,B and Appendix A). At the same time, we observed a significant decrease in IFN-λ expression in the lung tissues of SARS-CoV-2-infected patients (Figure 2C,D and Appendix A).

## 4. Discussion

In this study, we evaluated the expression of the immune–inflammatory mediators IL-1β, IL-6, IFN-β, and IFN-λ in postmortem lung specimens (i.e., the natural site of SARS-CoV-2 infection) of patients who died because of severe COVID-19 and control subjects. Interestingly we found that lung tissues of severe COVID-19 patients express higher levels of IL-1β and IFN- β and reduced levels of IFN-λ compared to controls. The levels of IL-6 were similar in the two groups.

Notably, the cytokines that we investigated are normally expressed in human tissue; in particular, as shown by our data, IL-1β, IL-6, and IFN-β are constitutively basally expressed in control cases, while IFN-λ is more expressed in control lung tissue and downregulated in COVID-19 patients’ lungs. To our knowledge, no studies report IFN-β and IFN-λ levels in lung specimens of deceased healthy donors, while a previous study showed IL-1β and IL-6 expression [18]. This latter study confirms our reported data on IL-1β, showing an increased expression in COVID-19 patients. As for IL-6, the reported data suggest an increased expression in COVID-19 patients, a finding that differs from what we described. However, it is important to underline that the COVID-19 cohorts analyzed in the two studies are different since our cohort includes only hospitalized patients, while the other group comprehends non-hospitalized patients. Coherently, another study conducted on postmortem lung specimens of hospitalized patients confirmed that IL-6 expression is not increased in COVID-19 patients’ lungs, which are the main site of infection leading to morbidity and mortality [8]. IL-1β is a proinflammatory cytokine that is rapidly expressed upon the activation of NLRP-3 inflammasome by PAMPs or DAMPs, and its expression is maintained during the inflammatory process [19]. Due to the various origin of PAMPs and DAMPs, is usual to have a basal expression of IL-1β in postmortem tissues. As for IL-6, it acts as a major proinflammatory mediator for the induction of the acute inflammatory response since it can induce the production of a variety of proinflammatory chemokines and cytokines, but its expression is progressively reduced [20]. Therefore, our findings that highlight equal IL-6 levels in lung tissue specimens of COVID-19 patients and control subjects might support the idea that IL-6 expression in the lungs could not be the leading cause of the COVID-19 proinflammatory cytokine storm.

Interferons, particularly type I and III interferons, are key molecules in the regulation of the innate immune response to viral infections. Type I IFN-β is mainly produced by epithelial cells, while type I IFN-α is mainly produced by circulating plasmacytoid dendritic cells [21]. While several studies reported delayed/impaired production of IFN-α in blood samples of COVID-19, the role of IFN-β in this scenario is mainly unexplored due to the almost undetectable levels of this molecule in blood samples [1,11,22]. Not surprisingly, given the pivotal role of this molecule in the antiviral responses, we found increased levels of IFN-β in lung specimens of patients who died of COVID-19 compared to patients who died of non-infectious conditions. The lack of a control group of patients who died because of other infection conditions outside of COVID-19 does not allow us to draw any conclusion on the magnitude and efficiency of this induction. However, the data show that significant IFN-β production occurs in the lungs of severe COVID-19 patients, which might reflect the weak clinical effect of exogenous administration of IF-β in severe COVID-19 patients [23].

By contrast, and in line with previous observations performed in blood samples [1,12,13], we found lower levels of IFN-λ in lung specimens of patients who died because of severe COVID-19 compared to the controls. In contrast, it has been reported that COVID-19 patient morbidity correlates with the high expression of IFN-type I and III in the bronchoalveolar lavage fluid (BALF), compared to subjects with other infectious or noninfectious lung pathologies [15]. However, as also suggested by Sposito and colleagues, IFN-λ production and release follow a specific timing, and the effective contribution of different cell types needs further investigation.

IFN-λ has pleiotropic activity, including not only antiviral effects but also modulatory effects on immune–inflammatory pathways [24]. It has been shown in animal models that IFN-λ can have anti-inflammatory effects via inhibition of neutrophil infiltration and suppression of proinflammatory cytokines such as IL-1β [25]. Interestingly, we found that the lower expression of IFN-λ in the lungs of patients who died because of COVID-19 compared to control was paralleled by a higher expression of IL-1β in the same group. Thus, the deranged innate immune response that can occur in severe COVID-19 patients might also contribute to amplifying the cytokine storm that follows the infection, indicating that a possible effective adjuvant therapy for the treatment of COVID-19-induced pneumonia could be the synergic inhibition of the proinflammatory IL-1β combined with the administration of IFN-λ [26].

## Figures and Tables

**Figure 1 pathogens-11-01390-f001:**
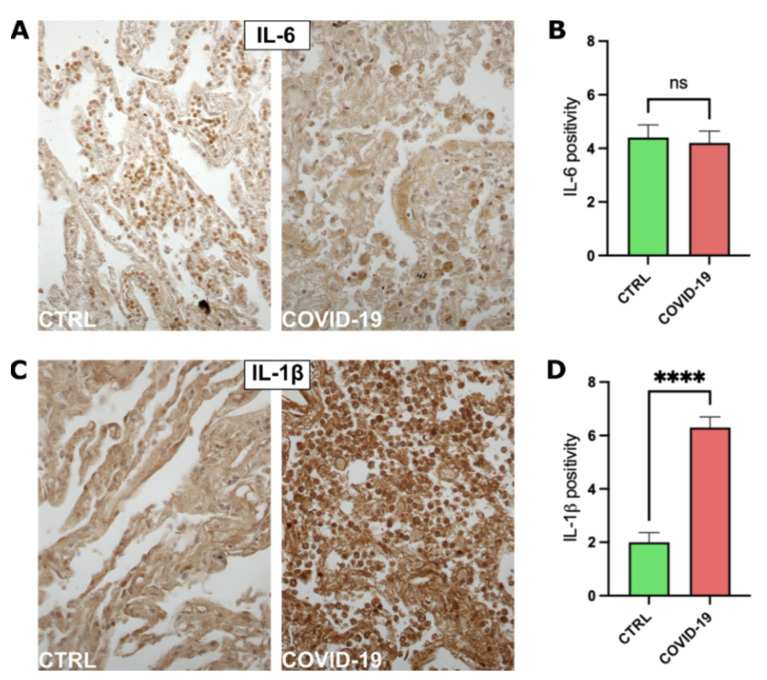
Immunohistochemical analysis for proinflammatory cytokine detection. (**A**) Representative images of lung specimens stained for IL-6. (**B**) Graph representing the semiquantitative analysis performed on IL-6-stained specimens, CTRL group *n* = 10, COVID-19 group *n* = 10. (**C**) Representative images of lung specimens stained for IL-1β. (**D**) Graph representing the semiquantitative analysis performed on IL-1β-stained specimens, CTRL group *n* = 10, COVID-19 group *n* = 10. ns = not significant, **** = *p* < 0.0001.

**Figure 2 pathogens-11-01390-f002:**
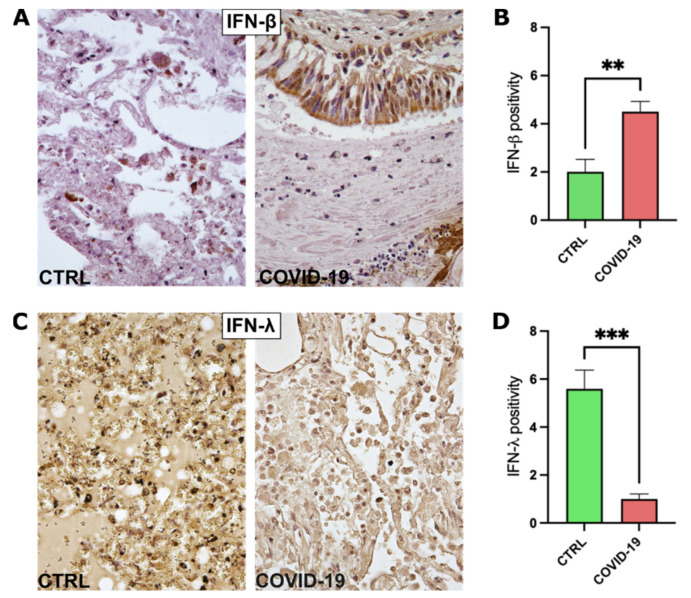
Immunohistochemical analysis for IFN detection. (**A**) Representative images of lung specimens stained for IFN-β. (**B**) Graph representing the semiquantitative analysis performed on IFN-β-stained specimens, CTRL group *n* = 10, COVID-19 group *n* = 10. (**C**) Representative images of lung specimens stained for IFN-λ. (**D**) Graph representing the semiquantitative analysis performed on IFN-λ-stained specimens, CTRL group *n* = 10, COVID-19 group *n* = 10. ** = *p* < 0.01, *** = *p* < 0.001.

## Data Availability

Not applicable.

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
