# Peer review of "SARS-CoV-2 Infection Prompts IL-1β-Mediated Inflammation and Reduces IFN-λ Expression in Human Lung Tissue"

_pathogens, 2022, doi:10.3390/pathogens11111390_

Round 1
Reviewer 1 Report
The study by Vezzani et al has looked at the inflammatory cytokine scenario post SARS-COV-2 infection in lung tissue. The study is of interest as it notifies levels of key cytokines at the site of infection which is lung. The manuscript can be improved by providing relevant references throughout the manuscript as well as more importantly by providing all of the relevant data.
Major Comments:
1. For the sake of transparency and better visualization, staining patterns of all 10 patients and 10 controls should be made available. That will improve the quality as well as value of the data. This can be provided as supplementary material but since the paper has only two figures this data should be added in to the main text.
2. Some cytokines show prominent signal in the controls too such as IFN-lambda. Is this the general pattern? This can be clarified in the manuscript if there is any precedence in the literature. To make the data clearer, staining controls such as unstained or isotype antibody or secondary antibody should be provided.
3. The scores for the intensity of immunopositivity evaluated by 2 different investigators should also be provided as a table format in addition to staining data as mentioned in point -1 above.
4. References of recent literature are lacking. Especially those evaluating IFN-lambda. There are reports which have shown reduced levels of IFN-lambda in the lung of patients. These reports should be acknowledged and discussed. Similarly relevant references for NLRP3, IL-1b, IL-6, IFN-b should also be discussed.
Minor Comments:
Line 22: aimed to the should be changed to aimed at
Line 121: wat should be changed to what
Sars-Cov-2 should be changed to SARS-CoV-2 throughout the manuscript.
Reviewer 2 Report
Vezzani et al. submitted a short report providing an interesting correlation between IL-1β, IFN-λ and Sars-CoV-2 infection in the lungs of deceased patients. The report is interesting, however the authors over-state and over-analyzed their results in the discussion. Discussion need to be carefully re-written to pay attention to these shortcomings. Some general English editing is also needed throughout the manuscript.
Major comments:
- Authors only provided Antibodies manufacturer's. However, antibody catalog numbers also need to be provided in the method section.
- Can the authors provide more information regarding tissue collection. In particular, how long after death were the tissue collected? How many different collection centres (eg. hospitals)?
- I would recommend the authors to provide as supplemental data all the pictures taken (and the grade associated) showing the difference they report. It is a very limited report and it would help provide wait to the claims made, especially in the context of the multiple overstatements made in the discussion.
- Lines 133-136: "In this study we tested the hypothesis that proinflammatory cytokines have a key role in promoting the development of Sars-CoV-2 induced pneumonia thus representing a promising therapeutic target, by estimating the expression of IL-6, IL-1β, IFN-β and IFN-λ in COVID-19 autoptic lung specimens".
The first line of the discussion is clearly an overstatement as authors did not test that. The only results presented is the correlation (or its absence) between IL-6, IL-1β, IFN-β and IFN-λ in lungs of deceased COVID-19 patients. Authors do not test the effect of these molecules on SARS-CoV-2 in any way shape or form and need to consequently tune down the language and instead provide an accurate report of their data.
- Lines 156-159: "Our data strongly agree with this latter finding, indicating that IFN-β expression is increased in patients deceased for COVID-19, thus implying the improbable efficacy of IFN-β administration as adjuvant treatment for this pathology."
This line of the discussion is another clear overstatement as authors did not show that. The only results presented is the presence of IFN-β in lungs of deceased COVID-19 patients which is normal in the case of an infection. There is no information relative to difference in IFN-β and patients outcome.
- Line 159 to 161: "Interestingly, we have highlighted the involvement of class III interferons, namely IFN-λ, as a possible pharmacological target to modulate Sars-CoV-2 induced inflammation. "
This sentence is another clear overstatement. Authors did not test for IFN-λ effect on SARS-CoV-2.
- New work has been published between SARS-CoV-2 and other inflammatory molecules such as IL8. I would suggest the authors to expand their discussion and incorporate some of these findings to draw comparison and highlight the relevance of their own findings. An expanded context would greatly benefit the manuscript and the readers.
Reviewer 3 Report
Vezzani et al have studied levels of four different cytokines (IL-1_β, IL-6, IFN-β and IFN-λ) in biopsies obtained from ten patients who died of COVID-19 and compared them with patients who died of traumatic injuries. The concentrations were measured semi-quantitatively. Patients with COVID-19 had significantly lower levels of IFN-λ, higher levels of IFN-β and IL-1β and similiar levels of IL-6 compared to deceased trauma victims.
Comments:
The findings are interesting and there is a need for data on cytokines in lung tissue from patients with COVID-19. The report offers important information on especially interferon lambda.
Language, introduction and discussion need improvements.
The language in the paper needs improvement. There are several miss-spellings, missing words and sentences that are sometimes hard to understand.
I recommend an introduction more focused on what is known about your cytokines in patients with COVID-19. This could be a relevant study for example doi: 10.1016/j.cell.2021.08.016. This paper supports your interferon lambda finding: doi: 10.1002/jmv.26993.
You state that type III is fundamental to control various viral respiratory tract infection. This is controversial, and for humans, not supported in references.
Line 50: You cite a study of the effect of type I and III interferons on SARS-CoV-2 in cell culture. Since it was published, several studies of interferon treatment in patients with COVID-19 have been published: https://doi.org/10.1016/S2213-2600(20)30566-X for peg-IFN-λ and the studies on IFN-β that you cite in the discussion.
Your aim is a bit confusing and ambitious for the study. “…COVID-19 deceased patients in order to confirm the hypothesis of using IL-1β inhibitors and IFN-λ enhancer as COVID-19 adjuvant therapy”. You can´t confirm this in your study. There are several studies on the use of IL-1 inhibitors and, at least one on, interferon lambda in patients with COVID-19.
Method
I would like to know a little bit more about the period when the patients died. Was it before or after vaccination was available? Which viral variant dominated? It would also be interesting to know all the ages of the covid-cases and some baseline characteristics in a table. Younger patients, for example a 30-year-old patient, is more likely to have an inborn error of interferon immunity. I recommend a supplementary table describing all patients.
Statistics
It seems that you have ordinal data on a scale 0-6 for cytokine concentration. Then you should use a non-parametric test such as Mann Whitney or explain why you chose a T test.
Discussion
I think you should comment something on your controls. Are they relevant? Did you expect differences in type I interferons in viral infected lung tissue compared to uninfected lung tissue? Is this increase in type I interferons something bad?
The downregulation of IFN-λ was interesting and the finding should be discussed further. This is a study which support your finding DOI: 10.1002/jmv.26993 and I am sure there are more.
This is a study with a similar finding regarding IL-6. PMID: 33082228.
Line 141-144. You did not measure IL-6 in serum. I don’t understand how this is an aspect beyond the cytokine storm.
Line 144-146. Is this a new hypothesis derived from this study? You only have data from lung tissues postmortem.
Round 2
Reviewer 1 Report
The authors have provided all of the requested data. The manuscript is significantly improved.
I would suggest the authors add the following reference and discuss
https://www.sciencedirect.com/science/article/pii/S0092867421009909
Author Response
We thank the Reviewer for helping us in improving our manuscript. As suggested we added the proposed reference and discussed it. Please see reference 15. The changes have been highlighted in yellow.
Reviewer 2 Report
Authors appropriately addressed my comments:
- The rework of the discussion and the intro improved the manuscript.
- Concerning the new figures: Authors should add them as supplemental materials. It would leave the number of figures and the text length of the main manuscript unchanged and provide the images if the readers are further interested.
Author Response
We thank the Reviewer for helping us in improving our manuscript. As suggested we have added the supplementary figures. Please find the references in the text highlighted in yellow.